# A Non-Thinning Forming Method with Improvement of Material Properties

**DOI:** 10.3390/ma16010407

**Published:** 2023-01-01

**Authors:** Yankuo Guo, Yongjun Shi, Feng Guo

**Affiliations:** 1School of Mechanical and Automotive Engineering, Qingdao University of Technology, No. 777, Jialingjiang Road, Huangdao District, Qingdao 266520, China; 2College of Mechanical & Electronic Engineering, China University of Petroleum, Qingdao 266580, China

**Keywords:** thermal stress forming, baffle pressure method, large bending angle, no-thinning

## Abstract

Thickness thinning is one of the processing defects that tend to occur in traditional stamping or mechanical bending of the plate and tube. In the field of high mechanical performance requirements (such as pressure vessels), the thinning phenomenon cannot be ignored. Thermal stress forming has excellent characteristics of forming without thinning, but the forming angle of this method is small, thus limiting the promotion and application of the process in the field of the form. To solve the problem, thermal stress forming with the baffle pressure method (BPM) is proposed. The coupled thermodynamic model of BPM is established, and the bending angle and deformation mechanism of the BPM are investigated. Lastly, the grain size and microhardness are measured and discussed. Results of the bending angle show that the proposed method can increase the bending angle by 57.71 times compared with the traditional method. The bending angle of BPM is determined by both the thermal buckling and the baffle, and the baffle plays a major role. The results of grain size and microhardness analysis show that the method refines the grain size, increases the material microhardness by 1.31 times and thickens the deformation zone by about 2.75%. In addition, the analytical equation of beam bending with laser as the heat source is given in this paper; this has some significance for further enrichment and development of the basic theory of beam thermoplastic bending.

## 1. Introduction

Bending parts of plate and tube have a wide range of applications in production and life, such as in automobiles, in instruments, in the military industry, in household appliances, etc. Plate and tube are often bent by stamping, mechanical bending and some special processing (such as multi-point forming, progressive forming, etc.). However, there is often a problem with thickness reduction during the forming process. This problem will directly affect the mechanical properties, such as the strength and stiffness of the part. Therefore, the thinning rate is often used as one of the important criteria in measuring the quality of bending parts [1,2]. With the increasing quality requirements of bending parts, thinning during the bending process should be avoided or solved.

Currently, some new techniques and methods [3,4,5] try to solve the problem of thinning in machining. However, there are problems such as complicated equipment and poor accuracy (with rebound), and the results are not satisfactory. Thermal stress forming is a special bending technique. This technique has advantages that traditional stamping (mechanical bending) cannot replace, such as a wide range of processing materials, no need for expensive molds and less material consumption. In addition, the great advantage of this technique is that the forming process does not thin [6]. However, the technique only relies on thermal stress forming and has the disadvantage of a small forming angle, which limits the application and development of this forming technology.

Due to the type of heat source, there are many forms of thermal stress forming, such as flame stress forming, high-frequency induction stress forming, laser stress forming, etc. In this work, laser stress forming (easy to automate, precise and concentrated heat energy, high precision [7,8]) and plate (more widely used than tube) are selected as the main research objects.

Different methods have been tried to obtain large bending angles for laser stress forming. One of the most common methods is through optimization of process parameters. Abedinzadeh et al. [9] conducted an experimental study of the laser bending process for non-alloy carbon steel plates. It was shown that the scanning speed had the most significant effect on the bending angle. In contrast, the spot diameter had the least effect on the bending angle. To obtain a large bending angle, a large laser power, small laser diameter and low scanning speed are recommended. Hussain et al. [10] studied the bending behavior of plates using laser irradiation of a titanium alloy (Ti–6Al–4V). This study showed that the bending angle decreases with increasing spot diameter and scanning speed. Nejad et al. [11] investigated the optimization of laser stress forming parameters using a combination of statistical methods and finite element modeling. The results showed that the bending angle increased with increasing laser power. The bending angle decreased with increasing laser diameter and scanning speed. Changing the shape of the laser spot is an effective method for increasing the bending angle. Shi [12] set up four laser heat source models for comparison experiments. The heat source models were a traditional circular spot, a square spot, a rectangular spot with an aspect ratio of 4:1 and a rectangular spot with an aspect ratio of 1:4. The results of the study showed that the rectangular 1:4 spot was 1.7 times the bending angle of the traditional circular spot. The study also concluded that the plastic strain difference and the width of the plastic zone play a decisive role in the amount of bending angle. Different cooling methods have a certain effect on the bending angle [13]. Kant et al. [14] set up two cooling methods (air cooling and gas cooling). Among them, gas cooling could increase certain bending angle values and produce martensitic organization in the heating zone. However, the gas cooling method would increase the equipment and more operations, which would affect the certain productivity [15]. Kalvettukaran et al. [16] proposed a simultaneous processing method of laser scanning (acting on the upper surface of the plate) and water cooling (acting on the lower surface of the plate). It was shown that this method could improve the bending angle by a factor of 1.2. The study concluded that the formation of a large temperature gradient on the upper and lower surfaces was the main mechanism to achieve the bending angle increase. Advance heat treatment of the material to be processed can increase the bending angle by a certain amount. Otsu et al. [17] conducted a laser thermal bending study on heat-treated materials and compared it with untreated materials. The results showed that the bending angle of the heat-treated material could be increased by about 1.5 times. In addition, the bending angle of the heat-treated material would increase rapidly with increasing laser power. Machining parameter optimization is a common method, but this method does not essentially improve the bending times. Different scanning methods have an effect on the bending angle. Safari [18] selected different scanning methods for a comparative study, which found that the bending angle of the linear method is significantly larger than that of the curved method. Although the papers [9,13,14] propose some new methods (i.e., changing the type of heat source, setting cooling conditions and material handling in advance), these methods improve the bending angle to a limited extent, and none of them exceeded 2 times (i.e., 1.7, 1.2 and 1.5 times). Therefore, the bending angle still has more room for improvement.

Thermal stress forming has the advantage of bending without thinning, but this forming has the disadvantage of low forming efficiency. To overcome this drawback, a method to increase the angle of thermal stress forming (BPM, Figure 1b) is proposed. A thermodynamic coupling model of BPM is developed to study the bending angle and deformation mechanism. Finally, the grain size, microhardness and the amount of thickening are observed. The results show that BPM can achieve efficient forming by thermal stress forming and effectively avoid the thinning problem that often occurs during the stamping process. This method can make thermal stress forming a high-quality process in the forming field.

## 2. Simulation and Experiment

In order to better analyze the forming mechanism, the finite element simulation approach is used. Among them, the temperature and deformation fields are established based on indirectly coupled nonlinear thermodynamic correlation theory.

### 2.1. Specific Steps of BPM

The specific process of BPM is as follows:Before laser scanning, the two ends of the plate are pressured with baffles to keep the plate under pressure (Figure 2a).When the laser beam scans the surface of the plate, the heating zone will undergo plastic deformation because of the coupling of thermal stress and pre-stress (Figure 2b).After the laser scan is complete, wait for the plate to cool to room temperature and then remove the baffle (Figure 2c,d).

Because it is necessary to apply baffles at both ends of the plate, this method is called the baffle pressure method (BPM).

### 2.2. Meshing and Parameters

Workbench simulation software and OLID45 cells (eight nodes with greater geometric nonlinearity and large deformation) are chosen. The heated zone mesh is encrypted, and the non-heated zone is sparse to improve the computational efficiency (Figure 3a). A bilinear hardening model is used.

For the application of pressure, as can be seen in Figure 3, the real situation is that the baffle is applied with force to the plate through *x*-directional displacement (Figure 3b,c), so the simulation can directly set Δ*x*-directional displacement for force application (Figure 3c). Since the experimentally applied Δ*x*-directional displacement values (<1 mm) are small, it is difficult to control (not easy to measure) through displacement application. Therefore, the experiments can be applied by force value *F* (which can be displayed) in Figure 3c. In this case, the relationship between *F* and the value of Δ*x*-directional displacement is shown in Figure 3c, and the specific values are shown in Figure 3d, where *E* is taken as the value 2.2 × 10^11^ Pa at room temperature (Figure 4).

The selected parameters are presented in Table 1.

### 2.3. Experimental Verification

The main instruments required for the experiments are the laser equipment, the infrared thermography (to measure temperature), the laser displacement sensor (to measure displacement) and the pressure application device (to apply pressure). Their parameters are shown in Table 2, Table 3 and Table 4. We designed the pressure application device. Its specifications are as follows: a force value ranging from 0 kN to 4.5 kN, an accuracy grade of C2–C3, a sensitivity of 2.0 ± 0.003 mv/V and a comprehensive error of 0.020% RO. The total processing time is 120 s. The heating time is approximately 0–30 s, and the cooling time is 30–120 s. The baffle is removed at *t* = 40 s. Figure 5 shows the experimental site.

Some typical parameters are selected for the experimental validation (Table 5). The bending angles are compared in Figure 6a (average error of 5.26%), and the temperature peaks are compared in Figure 6b (average error of 4.84%). The simulation results are accurate.

## 3. Results and Discussion

### 3.1. Bending Angle

#### 3.1.1. Deformation Mechanism

No. 1 (traditional method, *P* = 280 W, *v* = 1 mm·s^−1^, *d* = 8 mm, *h* = 0.8 mm, *q = /*) and No. 2 (BPM, *P* = 280 W, *v* = 1 mm·s^−1^, *d* = 8 mm, *h* = 0.8 mm, *q =* 1.5 × 10^7^ Pa) are chosen for the comparison of deformation mechanisms. As can be seen in Figure 7 and Figure 8, the comparison of *z*-direction deformation between the two methods is shown in Figure 8a. The traditional method is free end warping (Figure 8c), while BPM is intermediate warping (Figure 8d). The bending angle of the traditional method is 0.14 deg (*z*-directional deformation −0.13 × 10^−4^ m), and the bending angle obtained by BPM is 8.08 deg (*z*-directional deformation 1.06 × 10^−3^ m). BPM can improve the bending angle by 57.71 times.

For the traditional method, the deformation mechanism is based on the buckling mechanism. According to the buckling mechanism [20,21], the bending angle *θ_TB_* can be expressed by [22]
(1)$$\theta_{TB}={\left(\frac{36\alpha_{th}AP}{c\rho Evh^{2}}\right)}^{\frac{1}{3}}$$
where *A* is the laser absorption coefficient.

According to the knowledge of structural mechanics, the hinged bar is fixed at both ends when it is subjected to temperature load (Figure 9a). The bar undergoes thermal expansion, which will generate compressive stress inside the bar [23]. When the temperature rises to a certain number, the compression bar will be buckled and deformed. According to the paper [24], when the bar undergoes thermal buckling, the amount of deformation variation in the *z*-direction is
(2)$$z=\frac{2l}{\lambda }\sqrt{\alpha_{th}\Delta T{\left(\frac{\lambda }{\pi }\right)}^{2}-1}\sin\frac{\pi }{l}$$
where Δ*T* is the temperature rise value, and λ is the flexibility. BPM can be simplified to the problem of thermal buckling of a fixed hinged compression bar at both ends. When laser scanning is performed in the heating zone, according to the paper [25], the temperature rise Δ*T* in the heating zone can be expressed as follows.
(3)$$\Delta T=\frac{6.92AP}{\pi d^{2}k}\sqrt{\frac{\gamma d}{\pi v}}$$
where γ=k/ρc, γ is the thermal diffusivity. The heated area is expanded by heat, and the plate will undergo axial elongation (*x*-direction). Due to the blocking of the baffle, the plate will not be able to elongate. At this time, the plate will be subjected to a large coupling force (compressive stress and thermal stress). The bending moment will cause the upper and lower surfaces to be compressed and tensioned, respectively, thereby resulting in large bending angles. When the plate length is *l*, further derivation of Equation (2) will yield the bending angle *θ_B_* formed by the baffles as,
(4)$$\theta_{B}=2\arctan\left(\frac{z}{\frac{l}{2}}\right)=2\arctan\left(\frac{4}{\lambda }\sqrt{\alpha_{th}\Delta T{\left(\frac{\lambda }{\pi }\right)}^{2}-1}\sin\frac{\pi }{l}\right)$$

In addition, there is also a thermal buckling angle in BPM. Therefore, the bending angle *θ_TB_* formed by thermal buckling is also taken into account. In summary, the BPM bending angle will be obtained as follows:(5)$$\theta_{BPM}=\theta_{TB}+\theta_{B}={\left(\frac{36\alpha_{th}AP}{c\rho Evh^{2}}\right)}^{\frac{1}{3}}+2\arctan\left(\frac{4}{\lambda }\sqrt{\alpha_{th}\Delta T{\left(\frac{\lambda }{\pi }\right)}^{2}-1}\sin\frac{\pi }{l}\right)$$

By comparing Equations (1) and (5), it is obvious that the bending angle $\theta_{BPM}$ is greater than $\theta_{TB}$. In addition, the bending angle of the traditional method is determined by thermal buckling, and the BPM is formed by the joint action of the thermal buckling and baffle.

The above equations are simple qualitative theoretical derivation. For a more intuitive and quantitative comparison, the measurement point *P*_m_ is chosen at the center of the plate (Figure 8b) for comparison of the x-direction stress (Figure 8g,h). Figure 8g reveals that only region *s* (traditional method) satisfies the deformation conditions. In contrast, the plastic area of BPM is *s*_1_–*s*_2_ (Figure 8h). The area of *s*_1_–*s*_2_ is much larger than the area of *s*_._ Combined with Figure 8g,h, the value of $\theta_{B}$ in Equation (5) is much larger than the value of $\theta_{TB}$, so the baffle plays a major role in BPM.

Equation (5) is the qualitative analytical equation for beam bending with laser as the heat source. Due to the adoption of reasonable theoretical simplification, the equation has generality. Therefore, it can provide some reference for the plate bending with laser as the heat source and can enrich the basic theory of beam thermoplastic bending.

#### 3.1.2. Pressure and Plate Thickness

Figure 10a shows the bending angle of BPM compared with the traditional method in the case of different baffle pressures. From Figure 10a, it can be seen that the bending angle increases with the increase of baffle pressure. This is because an increase in baffle pressure means that the plate will be subjected to greater pressure (Figure 10b). When the thermal expansion of the plate occurs (the value of length *l* becomes larger), the plate will have more deformation in the *z*-direction Equation (2). The large *z*-directional deformation will create a larger bending angle.

Figure 11 depicts that the bending angles θ (both methods) show a decreasing trend as the thickness *h* increases. The bending angle of the plate is mainly affected by the resistive moment, which is mainly related to the thickness of the plate. The thicker the plate is, the larger the resistive moment of deformation is, and the smaller the bending angle is [26]. In addition, from Equation (1) (traditional method) and Equation (5) (BPM), it can be seen that the bending angle *θ* is inversely proportional to the plate thickness *h* in both methods. That is, the thicker the plate *h* is, the smaller the bending angle *θ* is.

#### 3.1.3. Processing Parameters

Figure 12a shows that, as the laser power *P* increases, the bending angle will increase. When the laser power increases, it will increase the temperature of the heated zone (Figure 12b). The high temperature will favor the formation of stress difference (Figure 8h). Eventually, a large bending angle is formed. An increase in the scanning speed will result in a decrease in the bending angle from Figure 12c. An increase in scanning speed will lead to a decrease in laser heating time. This will cause a lower temperature in the heated zone (Figure 12d), which will prevent the formation of a large bending angle. As can be seen in Figure 12e, an increase in laser diameter will result in a decrease in the bending angle. The increase in laser diameter will result in the laser not being able to focus. This will result in the laser not being able to develop a higher temperature (Figure 12f), which will result in a small bending angle.

### 3.2. Grain Size and Microhardness

The dimension of the samples is 10 mm in length and 5 mm in width. The chemical composition of the samples is shown on paper [27]. The grain size of the samples is observed by a metallographic microscope. The microhardness of the heating zone and substrate is measured by a microhardness tester. The super depth of field microscope is used to measure the thickness of the cross-section of the samples.

#### 3.2.1. Grain Size

Figure 13 shows the grain size of the cross-section under the two methods. Three zones—namely, the heated zone, the transition zone and the substrate zone—are observed from the heating zone to the substrate.

It can be seen that the grain size of BPM is smaller than that of the traditional method. When the laser heats the plate (heating zone), the heating zone will expand with heat. Due to the baffle effect (limiting the expansion), the whole plate of BPM will be in a strong extrusion and bending deformation. On the one hand, the extrusion and bending will destroy some grains and form dislocations. On the other hand, the extrusion and deformation can increase the stored energy in the metal, which in turn speeds up the nucleation rate of recrystallization [28,29]. Based on the above two reasons, BPM can refine the grains. Grain refinement can improve the overall physical properties of the part (e.g., yield strength, fatigue strength, impact toughness, etc.) [30]. Therefore, the use of BPM as a method can lead to improved material properties.

#### 3.2.2. Microhardness

Figure 14 compares the microhardness (heated zone) under the two methods. The BPM microhardness value in this figure (158 HV) is greater than that of the traditional method (121 HV), and BPM increases the microhardness value by a factor of 1.31. From Section 3.2.1, it is clear that the extrusion produced by BPM will facilitate grain refinement, and grain refinement facilitates the increase in microhardness values.

Figure 14c shows that the microhardness of the substrate (118 HV) of BPM is higher than that of the traditional method (102 HV). The microhardness of the substrate can be increased by 1.16 times because of the compression produced by BPM, which is equivalent to the compression of the whole plate. Compression can improve the yield strength and microhardness of the whole plate [31]. Therefore, BPM cannot only improve the microhardness of the heating zone but can also improve the microhardness of the whole steel plate.

#### 3.2.3. Thickness Variation

Figure 15 shows a comparison of the thickening at the heated zone with the two methods. There is almost no increase in thickness with the traditional method (Figure 15a). In contrast, BPM can thicken the heated zone by about 22 µm (822 − 800 = 22 µm), which is about 2.75% thicker (Figure 15b). Among them, the evolution curve of the thickening in the length direction is shown in Figure 15d. Compared with the traditional method, BPM can achieve a certain amount of thickness increase and thus a better no-thinning effect.

When the laser heats the heating zone, the temperature of the heating zone rises; thermal expansion occurs; and the material softens (the yield strength decreases). At this time, the non-heated zone exerts an extrusion effect on the heating zone. In the traditional method, because there is no axial force, the non-heating zone has a weak extrusion effect on the heating zone. Therefore, the stacking height resulting in the formation of the heating zone is small. In BPM, due to the blocking effect of the baffle, the non-heated zone has a certain extrusion effect on the heated zone, which will lead to a small amount of thickening accumulation in the heated zone.

In this work, the BPM of the plate is studied. In theory, BPM can also achieve the effect of tube bending without thinning. However, to bend the tube more directly and efficiently, the heat source can select the high-frequency induction coil matching the circular tube (as shown in Figure 15c).

## 4. Conclusions

In this paper, a method (BPM) is proposed. A coupled thermodynamic numerical model about BPM is developed. The bending angle and the deformation mechanism of BPM are investigated. In addition, the bending angles under different pressures, different plate thicknesses and different processing parameters are compared and discussed. Finally, the grain size, microhardness and thickness of the heated zone are observed.
Compared with the traditional method, BPM can improve the bending angle by 57.71 times. The bending angle can be effectively improved by BPM. The bending angle of the traditional method is determined by thermal buckling. In contrast, the BPM is determined by both thermal buckling and baffle, but the baffle plays a major role.In the BPM method, the bending angle will increase as the pressure and laser power increase. In contrast, the bending angle decreases with the increase of the plate thickness, scanning speed and laser diameter.Compared with the traditional method, BPM can refine the grain size and increase the microhardness by 1.31 times. BPM has a certain effect on improving material properties. In addition, BPM can thicken the heating zone by about 2.75%, which can obtain a better bending without thinning effect.The analytical equation of beam bending with laser as heat source is established, which further enriches and develops the basic theory of beam thermoplastic bending, and has certain reference and significance for other plate bending with laser as heat source.

## Figures and Tables

**Figure 1 materials-16-00407-f001:**
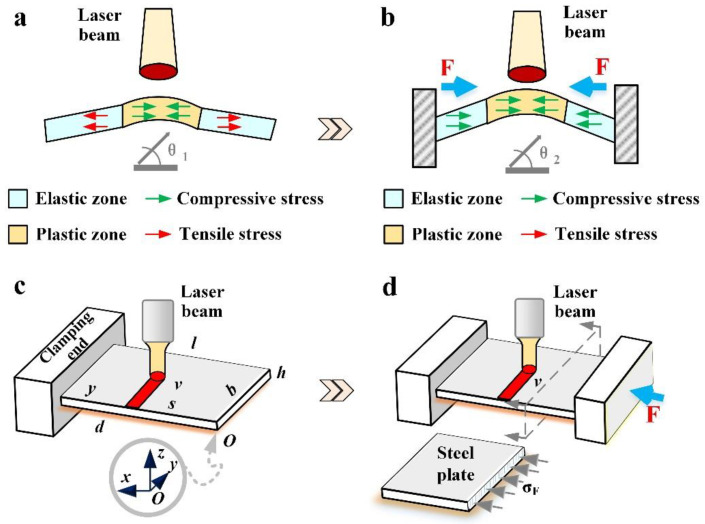
The comparison between the two methods: (**a**) traditional method (2D), (**b**) BPM (2D), (**c**) traditional method (3D) and (**d**) BPM (3D).

**Figure 2 materials-16-00407-f002:**
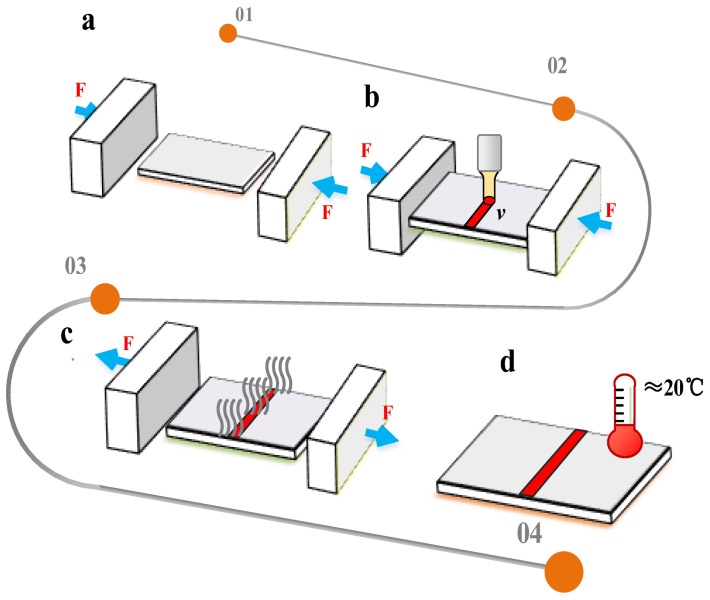
The specific process of BPM: (**a**) baffle application, (**b**) heating in progress, (**c**) removal of baffle and (**d**) cooling stage.

**Figure 3 materials-16-00407-f003:**
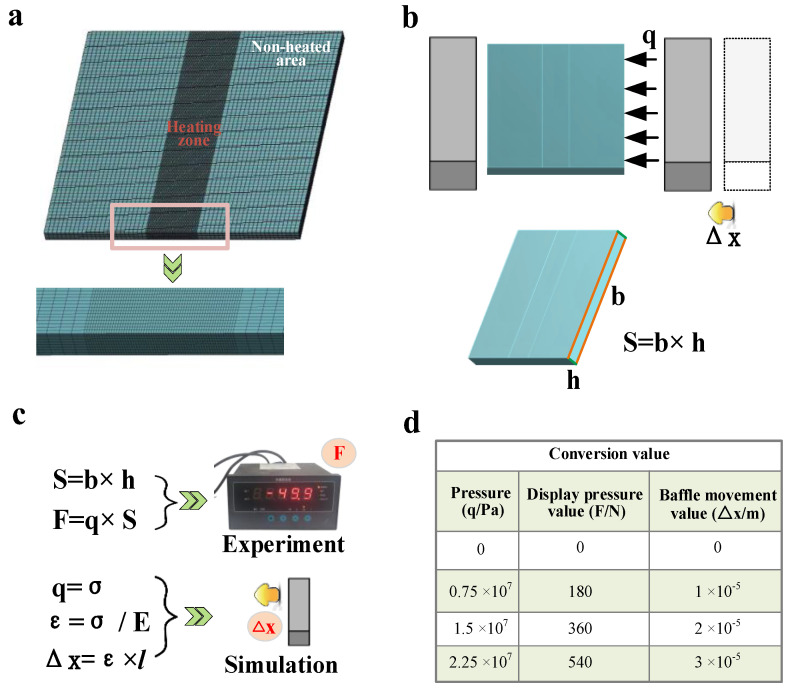
Schematic diagram of meshing, functional relationship between force and displacement. (**a**) Schematic diagram of the grid division, (**b**) schematic diagram of the action of the baffle, (**c**) equation of the relationship between force and x-directional displacement and (**d**) specific values of force and x-directional displacement.

**Figure 4 materials-16-00407-f004:**
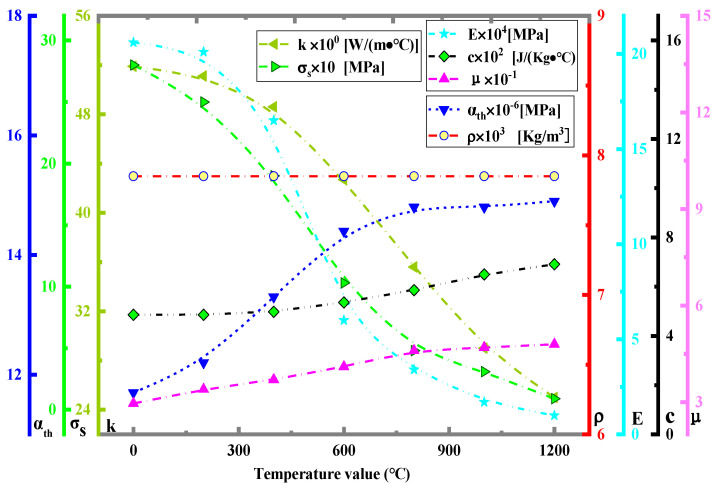
Thermophysical parameters at different temperatures.

**Figure 5 materials-16-00407-f005:**
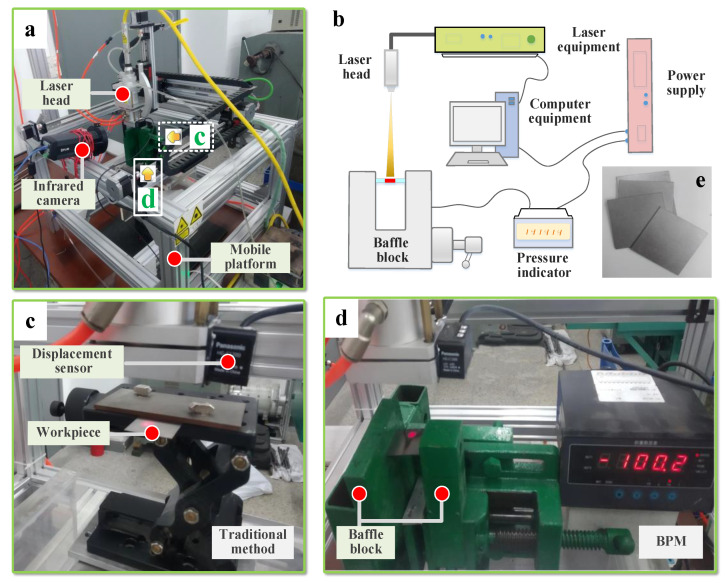
Experimental site: (**a**) overall diagram of the experimental equipment, (**b**) overall schematic of BPM, (**c**) the traditional method, (**d**) BPM and (**e**) experimental plate.

**Figure 6 materials-16-00407-f006:**
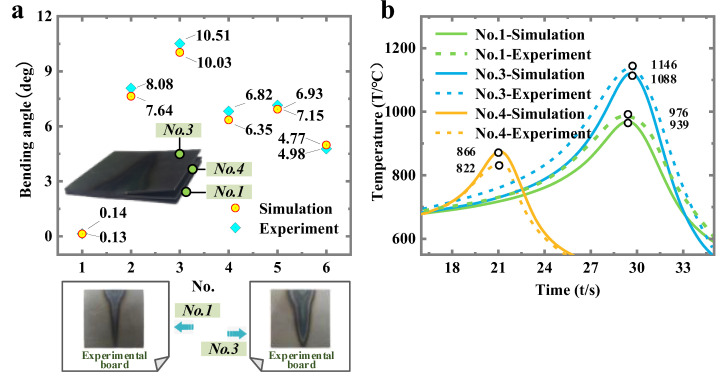
Comparison of results and the processed plate: (**a**) bending angle and (**b**) temperature peak.

**Figure 7 materials-16-00407-f007:**
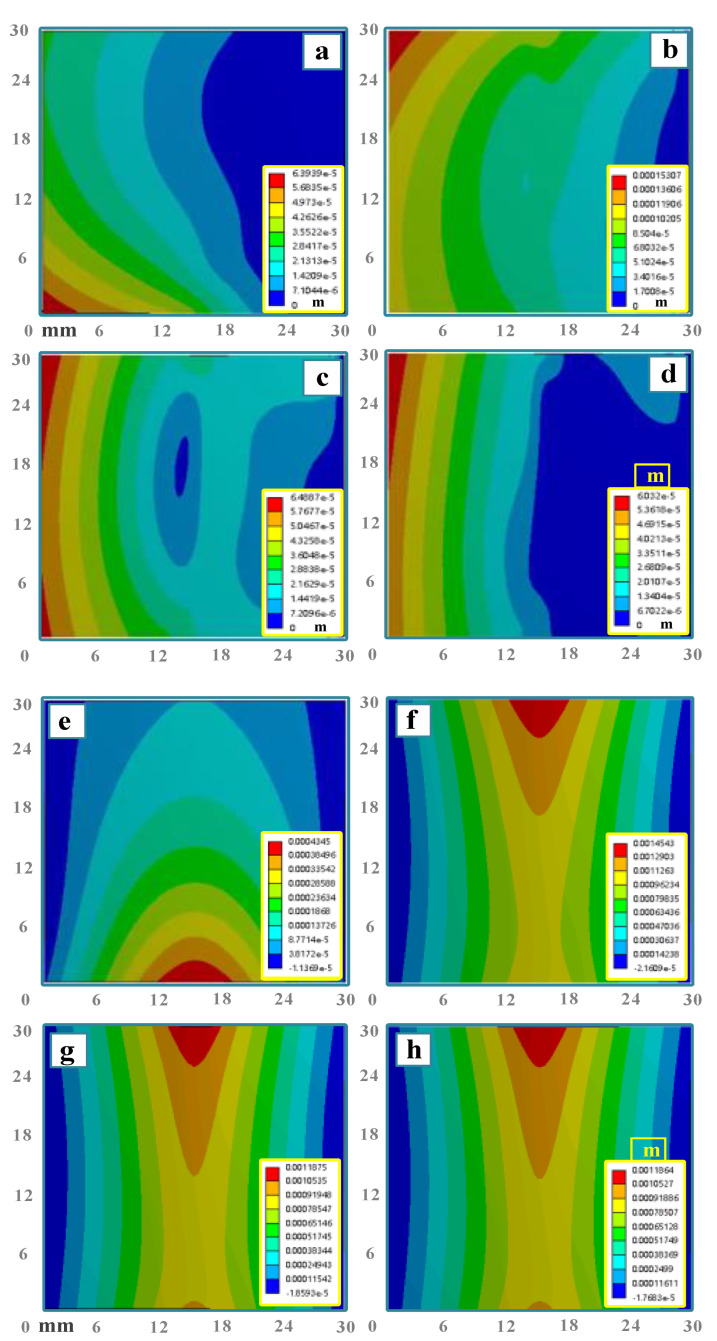
Cloud chart of *z*-direction displacement at different time: (**a**) *t* = 3 s (traditional method), (**b**) *t* = 33 s (traditional method), (**c**) *t* = 63 s (traditional method), (**d**) *t* = 93 s (traditional method), (**e**) *t* = 3 s (BPM), (**f**) *t* = 33 s (BPM), (**g**) *t* = 63 s (BPM) and (**h**) *t* = 93 s (BPM).

**Figure 8 materials-16-00407-f008:**
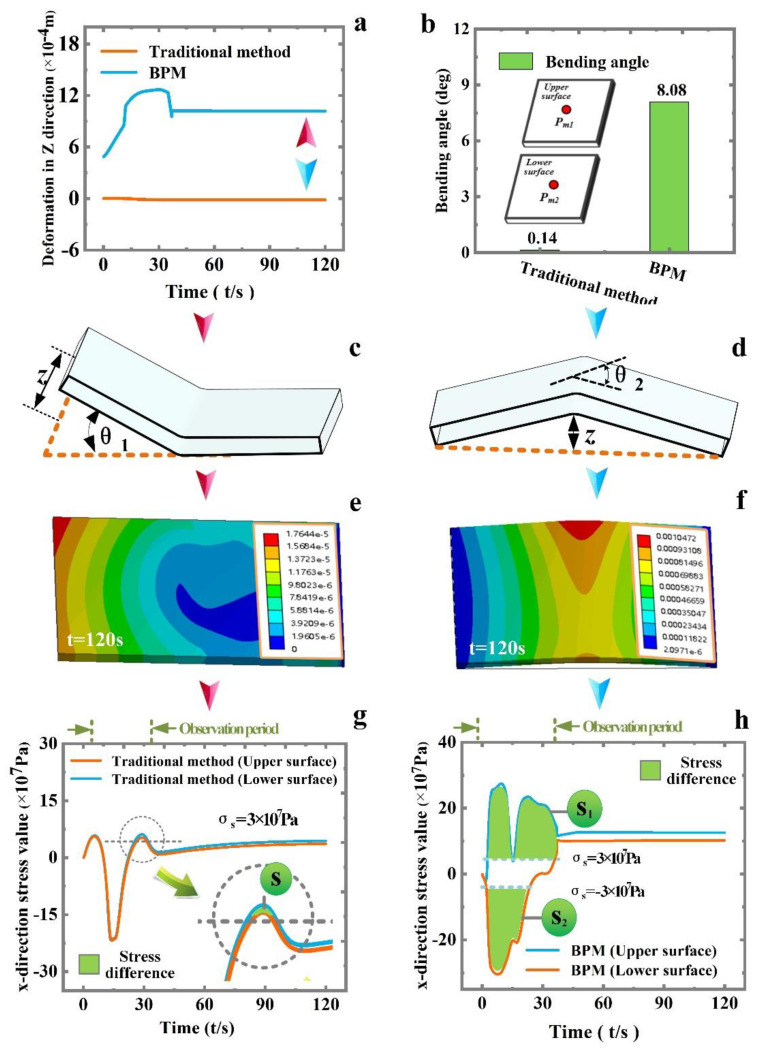
Comparison of traditional method and BPM. (**a**) z-directional deformation, (**b**) bending angle and measurement point, (**c**) bending form (traditional method), (**d**) bending form (BPM), (**e**) cloud diagram (traditional method), (**f**) cloud diagram (BPM), (**g**) stress value (traditional method) and (**h**) stress value (BPM).

**Figure 9 materials-16-00407-f009:**
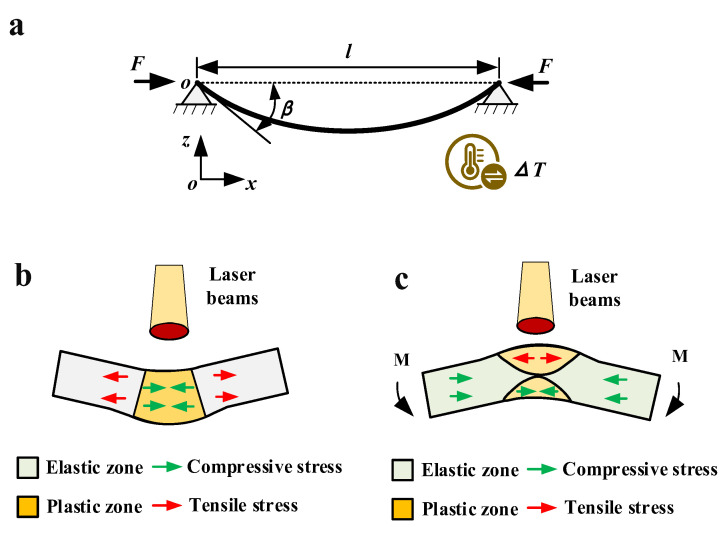
Schematic diagram of thermal buckling about the compression bars and the comparison of the bending deformation about two methods: (**a**) schematic diagram of thermal buckling of hinged compression bars fixed at both ends, (**b**) deformation diagram of the traditional method and (**c**) deformation diagram of BPM.

**Figure 10 materials-16-00407-f010:**
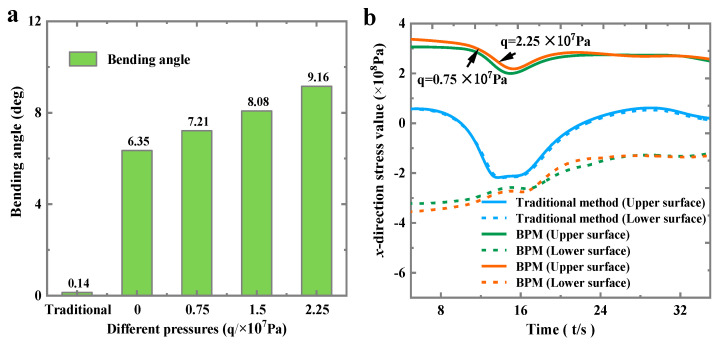
Comparison of stress between the two methods under different pressures: (**a**) bending angle and (**b**) stress value.

**Figure 11 materials-16-00407-f011:**
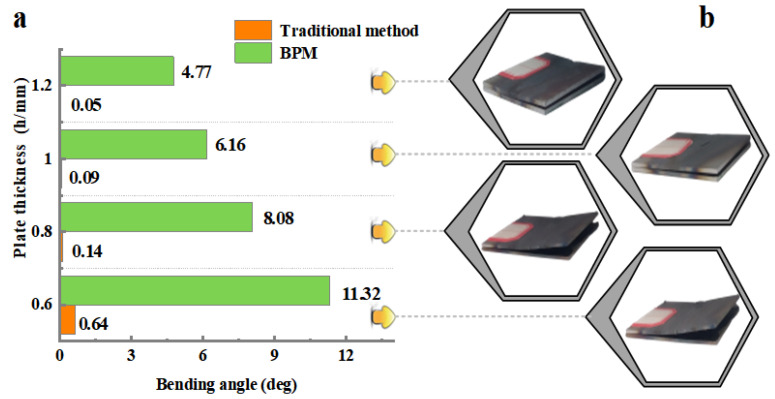
Comparison of the bending angle between the two methods under different plate thicknesses: (**a**) the bending angle and (**b**) the actual plate.

**Figure 12 materials-16-00407-f012:**
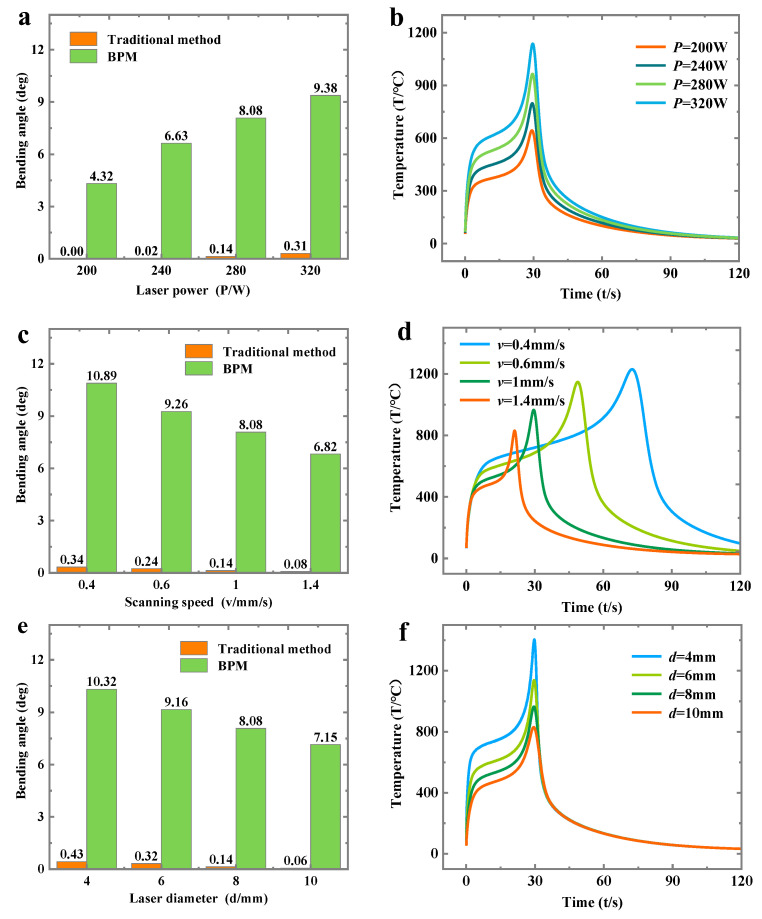
The comparison between the two methods under different processing parameters: (**a**) bending angle (different laser power), (**b**) temperature (different laser power), (**c**) bending angle (different scanning speed), (**d**) temperature (different scanning speed), (**e**) bending angle (different scanning diameter) and (**f**) temperature (different scanning diameter).

**Figure 13 materials-16-00407-f013:**
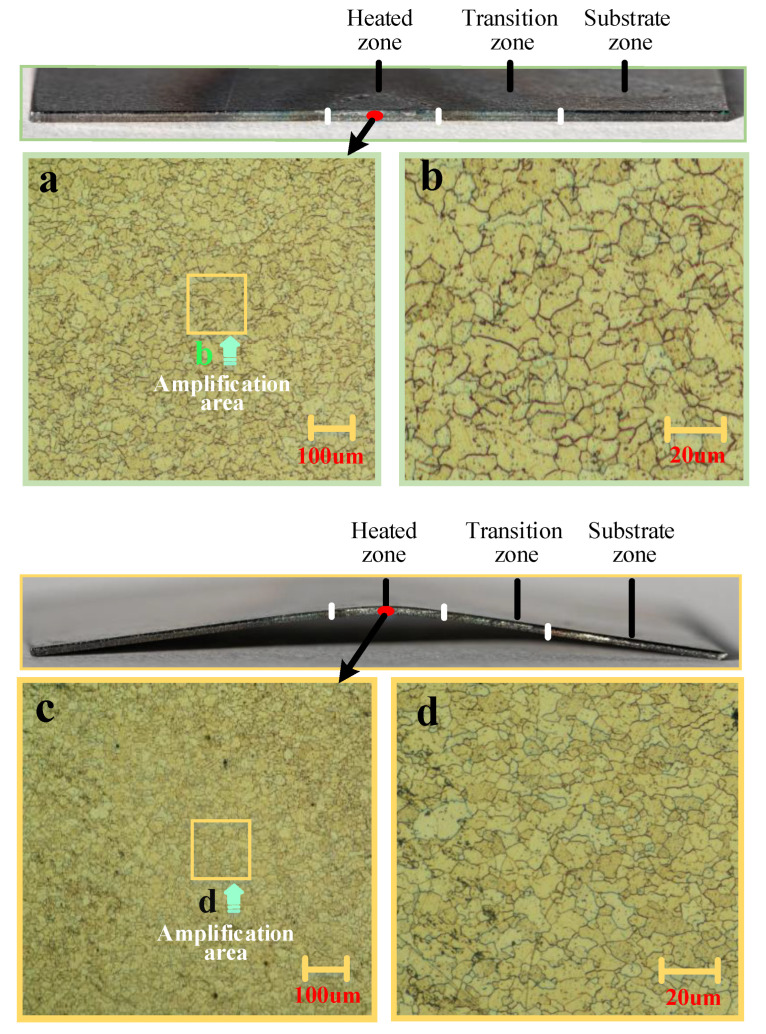
Comparison of grain size between the two methods: (**a**) ×100 time (traditional method), (**b**) ×500 time (traditional method), (**c**) ×100 time (BPM) and (**d**) ×500 time (BPM).

**Figure 14 materials-16-00407-f014:**
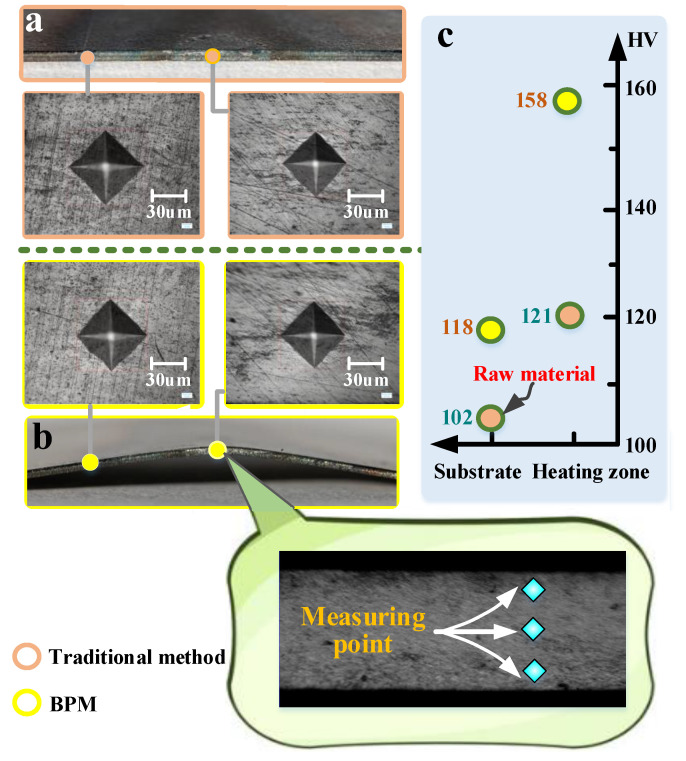
Comparison of microhardness values between the two methods: (**a**) traditional method, (**b**) BPM and (**c**) values of microhardness.

**Figure 15 materials-16-00407-f015:**
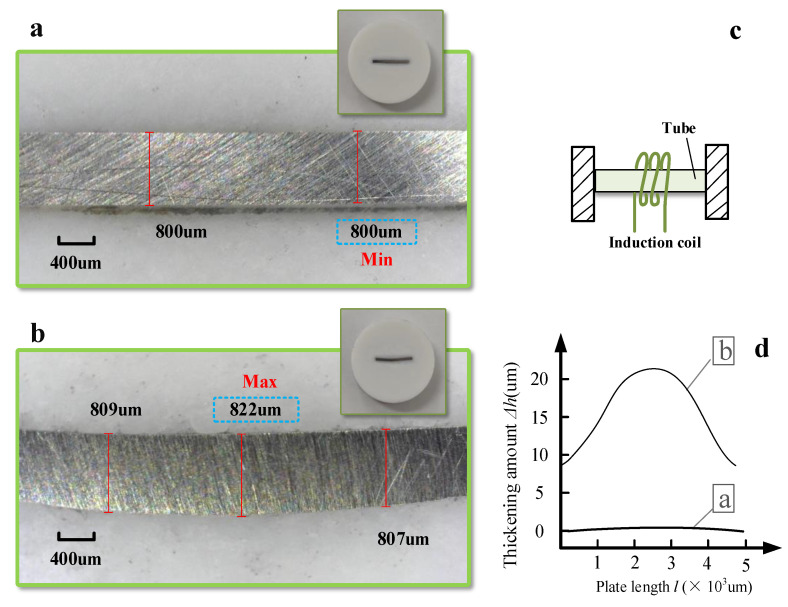
Comparison of thickening values between the two methods and tube forming: (**a**) traditional method, (**b**) BPM, (**c**) BPM for tube forming and (**d**) thickening curve comparison.

**Table 1 materials-16-00407-t001:** Processing parameters.

Case	Laser Power (*P*/W)	Scanning Speed (v/mm·s^−1^)	Spot Diameter (*d*/mm)	Plate Thickness (*h*/mm)	Scanning Position (*y*/mm)	Pressure (*q*/Pa)
Different plate thickness	280	1	8	0.6, 0.8, 1, 1.2	15	1.5 × 10^7^
Different pressure values	280	1	8	0.8	15	Traditional, 0.75 × 10^7^, 1.5 × 10^7^, 2.25 × 10^7^
Different laser power	200, 240, 280, 320	1	8	0.8	15	1.5 × 10^7^
Different scanning speed	280	0.6, 1, 1.4, 1.8	8	0.8	15	1.5 × 10^7^
Different spot diameters	280	1	7, 8, 9, 10	0.8	15	1.5 × 10^7^

Q235 steel is selected for simulation and experiment. This steel has moderate carbon content and good comprehensive performance. This material is widely used in various industries, such as ships, buildings, vehicles, etc. The thermophysical parameters of the material are changed according to the variation in temperature, and the relationship of the change is taken from paper [19] (Figure 4). The size of the plate (length *l* × width *b*) is 30 mm × 30 mm.

**Table 2 materials-16-00407-t002:** Parameters of the laser equipment.

Brand	Model	Frequency f (kHz)	Maximum Power (kW)	Central Wavelength
Chuangxin	MFSC-2000	0–50	2	1080 (nm)
Operation mode	Core diameter (µm)	Beam quality (mm × rad)	Cooling mode	
CW modulation	100	BPP = 2.8	Water-cooling	

**Table 3 materials-16-00407-t003:** Parameters of the infrared camera.

Brand	Model	Measurement Range (°C)	Time Response (ms)	Spatial Resolution (mrad)
Flir	A-615	−50~2000	8	0.68
Measurement error (%)				
±2				

**Table 4 materials-16-00407-t004:** Parameters of the displacement measuring instrument.

Brand	Model	Measurement Range (mm)
Panasonic	HG-C1400	±200

**Table 5 materials-16-00407-t005:** Typical parameters.

No.	Laser Power (*P*/W)	Scanning Speed (v/mm·s^−1^)	Spot Diameter (*d*/mm)	Plate Thickness (*h*/mm)	Pressure (*q*/Pa)
1	280	1	8	0.8	/
2	280	1	8	0.8	1.5 × 10^7^
3	320	1	8	0.8	1.5 × 10^7^
4	280	1.4	8	0.8	1.5 × 10^7^
5	280	1	10	0.8	1.5 × 10^7^
6	280	1	8	1.2	1.5 × 10^7^

## Data Availability

The results of the experimental tests are available upon request.

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
