# Peer review of "A Non-Thinning Forming Method with Improvement of Material Properties"

_materials, 2023, doi:10.3390/ma16010407_

Round 1

Reviewer 1 Report

The subject approached in the paper is new and of current interest.

A consistent analysis of the current stage is made and the principal achievements from the literature are presented in Introduction. The new method which combines thermal stress forming and mechanical compression is schematicaly presented.

In chapter2, authors present some elements used in their FEM simulations, the process parameters used in the experimental part, and the properties of the material used.

Chapter 3 is not clear. The chapter starts with the presentation of some results from FEM simulation. It is not clear how the authors made these simulations. The chapter continues with the presentation of an analytical model which is the base of their study. The experimental results presented in the chapter are clear and demonstrate the availability of their method.

The conclusions are clear and very well formulated.

The paper needs to be improved in connection with the FEM simulation. It will be better to eliminate the simulation part from your paper. If you decide to keep this part you must present in detail the elements of your FEM simulation and to make a comparison between the numerical and experimental results.

Author Response

Dear Editors and Reviewers:

Thank you for your letter and for the reviewers' comments concerning our manuscript entitled “ A non-thinning forming method with improvement of material properties” (ID: Materials-2015587). Those comments are all valuable and very helpful for revising and improving our paper, as well as the important guiding significance to our researches. We have studied comments carefully and have made correction which we hope meet with approval. Revised portion are marked in red in the paper. The main corrections in the paper and the responds to the reviewer's comments are as following:

Reviewer #1:

The subject approached in the paper is new and of current interest.

Response: Thank you very much for your approval. Your review of the paper was very clear & accurate. Also, thank you for your comments.

  1. Chapter 3 is not clear. The chapter starts with the presentation of some results from FEM simulation. It is not clear how the authors made these simulations. The chapter continues with the presentation of an analytical model which is the base of their study. The experimental results presented in the chapter are clear and demonstrate the availability of their method.The paper needs to be improved in connection with the FEM simulation. It will be better to eliminate the simulation part from your paper. If you decide to keep this part you must present in detail the elements of your FEM simulation and to make a comparison between the numerical and experimental results.

Response: I am very sorry that an explanation is needed here. For the sake of brevity of the paper, the simulation and experiment section was added as supplementary material. However, the supplementary material was not uploaded along with the manuscript, resulting in the absence of this section. This section has now been added to the manuscript (in the red section on pages 9-11) .

We appreciate for Editors/Reviewers’ warm work earnestly, and hope that the correction will meet with approval.

Once again, thank you very much for your comments and suggestions.

Yours sincerely,

Yankuo Guo

Reviewer 2 Report

With the aim of improving the formability, avoiding thinning and material damage during forming processes, the authors of this manuscript proposed the use of thermal stress forming, which avoids thinning in the bending process but has the limitation of reduced forming angle, with the baffle pressure method (BPM). By coupling these two methods, the authors proved that during the bending process, thinning was avoided and formability was improved considering the forming angle recorded during the tests. Good improvement results have been recorded compared to traditional technics.

In the opinion of the reviewer, the paper can be recommended for publication in materials Journal after revisions listed below:

- Abstract line 10, the word tube generally refers to pieces with a circular cross section, while the word plate refers to plates with a rectangular cross section.

- Most of the references reviewed in the introduction are old. However, it is best to add a few new references (2022) to see if someone else has already studied the topic.

- In the introductory section, authors can add a short paragraph on thinning reduction during hybrid incremental forming processes. New references have reviewed these processes, such as (DOI: 10.1177/09544089221093306).

- In section 2.1, it is best to draw a figure (your own figure) to recap the process with its different steps written in this section. If the process is the same of figure 1, 2 and 3 it is better to remove these figures from the introductory section.

- A figure presenting the FE model prepared should be added to the section 2.2

- A mesh description should be reported in this section as well.

- The mechanical proprieties of Q235 steel should be added in a table

- Authors should specify the type of hardening used in their simulation and if they used any damage model.

- The thickness of the plate is not specified.

- Figure 4. It is better to plot the spring-back by comparing the measured angle after springback and the target angle and make the comparison with the result of the traditional technic.

- To analyze the thinning phenomenon, it is better to draw the thinning evolution curve taking into account the length of the sheet. The authors should add this figure and make a comparison with the traditional technique.

Author Response

Dear Editors and Reviewers:

Thank you for your letter and for the reviewers' comments concerning our manuscript entitled “ A non-thinning forming method with improvement of material properties” (ID: Materials-2015587). Those comments are all valuable and very helpful for revising and improving our paper, as well as the important guiding significance to our researches. We have studied comments carefully and have made correction which we hope meet with approval. Revised portion are marked in red in the paper. The main corrections in the paper and the responds to the reviewer's comments are as following:

Reviewer #2:

With the aim of improving the formability, avoiding thinning and material damage during forming processes, the authors of this manuscript proposed the use of thermal stress forming, which avoids thinning in the bending process but has the limitation of reduced forming angle, with the baffle pressure method (BPM). By coupling these two methods, the authors proved that during the bending process, thinning was avoided and formability was improved considering the forming angle recorded during the tests. Good improvement results have been recorded compared to traditional technics.

Response: Thank you very much for your suggestions and comments. Through your comments, the quality of the paper has been greatly improved. In addition, these comments have deepened my understanding of paper writing. Finally, I would like to express my gratitude here.

I am very sorry that an explanation is needed here. For the sake of brevity of the paper, the simulation and experiment section was added as supplementary material. However, the supplementary material was not uploaded along with the manuscript, resulting in the absence of this section. This section has now been added to the manuscript (in the red section on pages 9-11) .

  1. Abstract line 10, the word tube generally refers to pieces with a circular cross section, while the word plate refers to plates with a rectangular cross section.

Response: The main purpose here is to emphasize that this method can also be used for tube forming. To avoid misunderstandings, the description of the tube part has been modified (in the red section on pages 1-2).

  1. Most of the references reviewed in the introduction are old. However, it is best to add a few new references (2022) to see if someone else has already studied the topic.

Response: The original two older papers are removed and five papers from 2022 are added [3][5][10][11][18].

  1. In the introductory section, authors can add a short paragraph on thinning reduction during hybrid incremental forming processes. New references have reviewed these processes, such as (DOI: 10.1177/09544089221093306).

Response: This section has been added (red section on page 2). Thank you very much for the information you have provided.

  1. In section 2.1, it is best to draw a figure (your own figure) to recap the process with its different steps written in this section. If the process is the same of figure 1, 2 and 3 it is better to remove these figures from the introductory section.

Response: The process diagram in Fig. 1 is removed and a new process diagram is redrawn (Fig. 2).

  1. A figure presenting the FE model prepared should be added to the section 2.2

Response: The diagram of the FE model is added (Fig. 3).

  1. A mesh description should be reported in this section as well.

Response: The mesh description has been added (Fig.3a and the red section on page 6 ).

  1. The mechanical proprieties of Q235 steel should be added in a table

Response: Fig. 4 shows the mechanical properties of the material. It is shown in the form of curves and not in the form of a table.

  1. Authors should specify the type of hardening used in their simulation and if they used any damage model.

Response: A bilinear hardening model is used and is explained in the paper (the red section on page 6 ). Since the degree of deformation in this method is not very large. The thermal form (with softening effect and not easy to rupture) is also used. Therefore, the type of damage is not considered.

  1. The thickness of the plate is not specified.

Response: The thickness dimensions are added in Table 5.

  1. Figure 4. It is better to plot the spring-back by comparing the measured angle after springback and the target angle and make the comparison with the result of the traditional technic.

Response: There is no preload applied in this method, so the springback will not be significant. Also, with the thermoformed form, the springback will be smaller.

  1. To analyze the thinning phenomenon, it is better to draw the thinning evolution curve taking into account the length of the sheet. The authors should add this figure and make a comparison with the traditional technique.

Response: Contrast curves have been added in Fig. 15d (in the red section on page 22).

We appreciate for Editors/Reviewers’ warm work earnestly, and hope that the correction will meet with approval.

Once again, thank you very much for your comments and suggestions.

Yours sincerely,

Yankuo Guo

Round 2

Reviewer 1 Report

Thank you for your comments and actions to improve the paper.

The paper could be published in this form.